# From Molecular Insights to Clinical Management of Gestational Diabetes Mellitus—A Narrative Review

**DOI:** 10.3390/ijms26178719

**Published:** 2025-09-07

**Authors:** Mohamed-Zakaria Assani, Lidia Boldeanu, Maria-Magdalena Manolea, Mihail Virgil Boldeanu, Isabela Siloși, Alexandru-Dan Assani, Constantin-Cristian Văduva, Anda Lorena Dijmărescu

**Affiliations:** 1Doctoral School, University of Medicine and Pharmacy of Craiova, 200349 Craiova, Romania; mohamed.assani@umfcv.ro (M.-Z.A.); alexandruassani@gmail.com (A.-D.A.); 2Department of Immunology, Faculty of Medicine, University of Medicine and Pharmacy of Craiova, 200349 Craiova, Romania; mihail.boldeanu@umfcv.ro; 3Department of Microbiology, Faculty of Medicine, University of Medicine and Pharmacy of Craiova, 200349 Craiova, Romania; lidia.boldeanu@umfcv.ro; 4Department of Obstetrics and Gynecology, Faculty of Medicine, University of Medicine and Pharmacy of Craiova, 200349 Craiova, Romania; cristian.vaduva@umfcv.ro (C.-C.V.); lorenadijmarescu@yahoo.com (A.L.D.)

**Keywords:** gestational diabetes mellitus, insulin resistance, placental biomarkers, epigenetics, adipokines

## Abstract

Gestational diabetes mellitus (GDM) is one of the most common metabolic complications during pregnancy, affecting up to 14% of pregnancies globally. GDM is characterized by glucose intolerance that arises or is first identified during pregnancy and is linked to significant short- and long-term adverse outcomes for both mothers and their offspring. The pathophysiology of GDM involves more than maternal insulin resistance and β-cell dysfunction. It is influenced by complex interactions among placental hormones, adipokines, inflammatory mediators, and oxidative stress pathways. Additionally, placental-derived exosomes and metabolomic signatures have emerged as promising biomarkers for early prediction and monitoring of the disease. Despite advancements in clinical diagnosis and management, including lifestyle interventions and pharmacological treatments, current strategies are still inadequate to prevent complications for both mothers and newborns entirely. Recent molecular insights into GDM development have been explored, along with emerging biomarkers and potential therapies. This synthesis also considers prospects for precision medicine strategies that could significantly improve GDM management. The urgent need for improved prevention and treatment of GDM is evident. A deeper understanding of the molecular foundations of GDM is essential and urgent, as it may enhance clinical outcomes and provide opportunities for early prevention of intergenerational metabolic disease risk.

## 1. Introduction

Gestational diabetes mellitus (GDM) is defined as any degree of glucose intolerance first diagnosed during pregnancy, typically in the second or third trimester, when prior overt diabetes has been ruled out [1,2,3]. Diagnostic standards vary, but the widely accepted International Association of Diabetes and Pregnancy Study Groups (IADPSG) and World Health Organization criteria recommend a 75 g oral glucose tolerance test (OGTT) with thresholds of fasting ≥ 5.1 mmol/L, 1 h ≥ 10.0 mmol/L, and 2 h ≥ 8.5 mmol/L plasma glucose [4,5,6]. In some regions, a two-step strategy (screening followed by OGTT) is still used for pragmatic reasons [7]. Globally, the prevalence of GDM is rising, with current estimates around 14% of pregnancies, depending on diagnostic criteria and population [8,9,10,11].

Key risk factors for GDM include maternal obesity, advanced maternal age (>35 years), family history of type 2 diabetes, polycystic ovarian syndrome (PCOS), previous macrosomic infant or previous GDM, and specific ethnic backgrounds (e.g., South Asian, African, Hispanic, Pacific Islander) [7,9,12]. Genetic predisposition also plays a role: polymorphisms in genes such as transcription factor 7-like 2 (*TCF7L2*), melatonin receptor 1B protein (*MTNR1B*), insulin receptor substrate 1 (*IRS1*), potassium voltage-gated channel subfamily Q member 1 (*KCNQ1*), among others, have been associated with impaired β-cell function, insulin secretion, and predisposition to GDM [13,14,15]. Lifestyle and environmental factors such as sedentary behavior, unhealthy diet, and obesity exacerbate insulin resistance during pregnancy [1,2,12,16].

Pregnancy naturally induces a state of progressive insulin resistance, especially in the second and third trimesters, mediated by placental hormones (human placental lactogen (hPL)), prolactin, progesterone, cortisol, estradiol, cytokines, and adipokines (e.g., tumor necrosis factor α (TNF-α), leptin, resistin) [2,9]. In healthy pregnancies, pancreatic β-cells expand and increase insulin output to maintain euglycemia. In GDM, β-cell compensation fails—often due to genetic defects, weight gain of the mother, and hormonal disorders, leading to maternal hyperglycemia [17,18,19]. Figure 1 illustrates the main molecular mechanisms involved in GDM.

These maladaptations link deeply to molecular mechanisms: oxidative stress and mitochondrial dysfunction in β-cells and placenta; aberrant hepatocyte growth factor/c-mesenchymal–epithelial transition factor (HGF/c-MET) signaling affecting β-cell proliferation; inflammatory activation involving macrophages and cytokines; and impaired insulin signaling pathways [22,23,24,25]. Epigenetic changes, including deoxyribonucleic acid (DNA) methylation, histone modification, and non-coding ribonucleic acids (RNAs) such as microRNAs, modify gene expression in maternal tissues and the placenta, influencing both maternal metabolism and fetal programming [8,9,26].

The maternal and fetal complications of GDM are considerable: mothers face higher rates of preeclampsia, hypertension, cesarean delivery, and birth trauma; infants face macrosomia, neonatal hypoglycemia, hyperbilirubinemia, respiratory distress, and increased neonatal intensive care unit (NICU) admissions [8,9]. Untreated or poorly controlled GDM further elevates these risks, including stillbirth and birth injuries [8,27]. It has recently been demonstrated that severe acute respiratory syndrome coronavirus 2 (SARS-CoV-2) severe infection during pregnancy is associated with higher odds of developing GDM [28].

Crucially, GDM has long-term sequelae: women with GDM presented higher chances to develop type 2 diabetes postpartum, especially if they required insulin or had obesity [29,30]. Offspring born to GDM mothers have higher lifetime risks for childhood obesity, impaired glucose tolerance, type 2 diabetes, hypertension, and cardiovascular disease—often attributable to fetal metabolic programming through epigenetic and microbiome alterations [3,8,31].

## 2. Pathophysiology and Molecular Mechanisms

### 2.1. β-Cell Dysfunction, Inflammation and Immune Regulation, Oxidative Stress and Mitochondrial Dysfunction

In healthy pregnancies, β-cells expand and increase insulin secretion to compensate for pregnancy-induced insulin resistance. In GDM, this compensation fails due to molecular stressors such as oxidative stress, glucolipotoxicity, and endoplasmic reticulum (ER) stress. Accumulation of reactive oxygen species (ROS) damages β-cells, impairing insulin gene transcription and secretion, and promoting apoptosis [18,32]. Additionally, failure of translational control and autophagy pathways further compromises β-cell survival and function [2,32]. Heterogeneity analyses show women with combined insulin resistance and β-cell dysfunction, compared to single defects, experience more severe GDM phenotypes and worse perinatal outcomes [33].

Systemic inflammation is central in promoting insulin resistance. Elevated TNF-α and IL-6 levels contribute to metabolic dysregulation. This inflammation radiates from adipose tissue into the liver, skeletal muscle, and placenta, aggravating insulin resistance across multiple tissues [2,34,35].

Recent studies in GDM placental and metabolic tissues demonstrate that mitochondrial dysfunction impairs adenosine triphosphate (ATP) generation and elevates ROS, which drives serine phosphorylation of *IRS1*, impairing insulin signaling across skeletal muscle, adipose and trophoblast cells. This reinforces a vicious cycle of oxidative damage, inflammation, and persistent insulin resistance [2,36,37]. Further mechanistic parallels are observed in type 2 diabetes and insulin resistance (IR) models, reinforcing the centrality of dysfunctional oxidative metabolism in promoting insulin resistance [38,39].

### 2.2. Insulin Resistance During Pregnancy, Adipokines and Metabolic Regulation

During pregnancy, rising levels of placental hormones, including hPL, leptin, TNF-α, interleukin 6 (IL-6), and cortisol, disrupt insulin signaling. hPL promotes lipolysis and increases free fatty acids, which activate protein kinase C (PKC) and induce serine phosphorylation of *IRS-1/2* rather than activation-promoting tyrosine phosphorylation. This impairs phosphatidylinositol 3-kinase (PI3K)/protein kinase B (PI3K/Akt) signaling and glucose transporter type 4 (GLUT4)-mediated glucose uptake in maternal tissues [2,20,40,41]. Leptin, often elevated in GDM, inhibits insulin receptor expression through janus kinase/signal transducer and activator of transcription (JAK/STAT) pathways, while hypoadiponectinemia reduces adenosine monophosphate (AMP)-activated protein kinase (AMPK) activity, further impairing GLUT4 activation and insulin sensitivity [10,42,43]. Chronic low-grade inflammation—driven by TNF-α and IL-6 from adipose tissue macrophages—activates stress kinases Jun N-terminal Kinase (JNK) and I-kappa-B Kinase (IKK), promoting *IRS1* serine phosphorylation and exacerbating systemic insulin resistance [2]. Leptin, an adipokine mainly secreted by adipose tissue and the placenta, is critical in regulating energy balance, insulin sensitivity, and metabolic adaptation during pregnancy. In women with GDM, circulating leptin levels are often elevated compared to healthy pregnancies, reflecting increased adiposity, placental overproduction, and a state of leptin resistance. This dysregulation is thought to contribute to the development of insulin resistance, impaired β-cell compensation, and adverse metabolic outcomes. Clinical studies have confirmed that higher maternal leptin concentrations, particularly in early pregnancy, are associated with an increased risk of developing GDM later in gestation [10,44].

Adiponectin, an adipokine secreted mainly by adipose tissue, exerts insulin-sensitizing, anti-inflammatory, and anti-atherogenic effects. Unlike leptin and visfatin, circulating adiponectin levels are typically reduced in women with GDM compared to healthy pregnancies. Low adiponectin concentrations, especially in early pregnancy, have been consistently identified as a predictor of subsequent GDM development, reflecting impaired insulin sensitivity and β-cell dysfunction. Furthermore, hypoadiponectinemia in GDM has been associated with greater maternal insulin resistance and increased risk of adverse metabolic outcomes in offspring. These findings suggest that adiponectin plays a protective role in pregnancy metabolism, and its deficiency contributes to the pathophysiology of GDM [45,46,47]. Cytokines such as TNF-α and IL-6, in GDM, lower adiponectin levels, disrupting adiponectin receptor 1 (AdipoR1) and adiponectin receptor 2 (AdipoR2) signaling, and reducing AMPK activation. This disruption impairs fatty acid oxidation and glucose uptake, worsening insulin resistance [2].

Resistin, another adipokine elevated in obesity, has also been implicated in insulin resistance and inflammatory responses, although its exact role in human GDM remains under investigation [48,49]. In the context of GDM, several studies report that maternal circulating resistin levels are dysregulated, often showing higher concentrations compared to normoglycemic pregnancies. This imbalance may contribute to the pro-inflammatory state and impaired insulin sensitivity characteristic of GDM. Moreover, resistin has been detected in placental tissue, where its expression may further influence maternal glucose metabolism and fetal nutrient supply. Overall, current evidence suggests that resistin plays a role in the pathophysiology of GDM by linking inflammation, adipose tissue dysfunction, and insulin resistance [48,50].

Visfatin is an adipokine with insulin-mimetic effects that is secreted by visceral adipose tissue and the placenta, and it participates in the regulation of glucose and lipid metabolism. Several original studies have reported that maternal circulating visfatin levels are altered in pregnancies complicated by GDM, commonly showing elevated concentrations in early and mid-gestation among women who develop GDM [50,51,52]. However, some investigations, particularly in later pregnancy, have found lower visfatin levels in women with GDM compared to normoglycemic controls, and visfatin levels have been inversely correlated with insulin resistance markers such as homeostatic model assessment of insulin resistance (HOMA-IR) in healthy pregnancies but not consistently in GDM. Overall, while the direction varies by timing and population, visfatin appears to be dysregulated in GDM and to have a potential role in insulin resistance and metabolic adaptation during pregnancy [53].

### 2.3. Epigenetic and Genetic Factors: MicroRNAs

Genetic variants in key genes, such as *TCF7L2*, *IRS1*, solute carrier family 30 member 8 (*SLC30A8*), cyclin-dependent kinase 5 regulatory subunit associated protein 1-like 1 (*CDKAL1*), *MTNR1B*, and glucokinase (*GCK*), have been linked to susceptibility to GDM through effects on β-cell function and insulin action. Epigenetic mechanisms, such as DNA methylation in metabolic gene promoters, histone modifications, and dysregulated microRNAs, modulate gene expression relevant to insulin signaling and glucose metabolism. These epigenetic marks may reflect in utero environment or maternal metabolic state and contribute to both GDM and fetal programming [40,54,55,56].

Specific microRNAs, such as miR-375, are highly enriched in pancreatic β-cells and play crucial roles in regulating insulin secretion and β-cell mass maintenance. Although predominantly examined within the context of type 2 diabetes mellitus, emerging evidence suggests that miR-375 may also be integral to β-cell adaptation processes during pregnancy and gestational diabetes mellitus (GDM). Additionally, other microRNAs, including miR-21 and miR-23a, have been identified through recent biomarker studies in GDM, implicating them in underlying mechanistic pathways of the disease [57,58,59,60,61]. miRNAs such as miR-21, miR-23a, miR-130a, miR-193a are consistently dysregulated in circulation and placental tissues from women who develop GDM, even in early pregnancy [62,63,64]. These exosomal miRNAs, derived from placental or maternal tissues, serve as both mechanistic mediators and early predictors, with candidate panels under active investigation [63].

GDM arises from a multifactorial network of molecular insults: placental and adipose-derived hormones triggering insulin resistance, failing β-cell compensation under oxidative and inflammatory stress, dysregulation of adipokines, and contributions from genetic susceptibility coupled with epigenetic modulation. These pathways converge on disrupted IRS-PI3K-Akt-GLUT4 signaling, mitochondrial impairment, stress-kinase activation, and defective insulin secretion [2,65,66].

Table 1 summarizes all the above pathophysiological processes and molecular mechanisms.

## 3. Biomarkers of GDM

Current screening approaches, mainly based on fasting plasma glucose (FPG), OGTT, and hemoglobin A1c (HbA1c), often detect GDM after pathological processes have already been established. This delayed diagnosis limits opportunities for early intervention, increasing the risk of adverse pregnancy outcomes. Therefore, the identification of biomarkers capable of predicting GDM in the first trimester or even preconceptionally has become a central research focus [9].

Biomarkers serve not only as diagnostic and predictive tools but also as windows into the molecular mechanisms underlying GDM. Advances in molecular biology and omics technologies—such as genomics, metabolomics, and proteomics—have revealed novel classes of biomarkers including circulating microRNAs, placental-derived exosomes, and metabolic signatures that reflect impaired insulin action, β-cell stress, inflammation, and oxidative imbalance. These emerging candidates carry dual potential: they may allow early detection and risk stratification while also identifying new therapeutic targets [69,70,71]. For a better overview, refer to Table 2.

Importantly, biomarker research in GDM is not limited to glucose metabolism alone. Increasingly, focus has shifted toward integrative biomarker panels that combine clinical characteristics (e.g., maternal age, BMI, family history) with molecular indicators of immunometabolism, adipokine signaling, and placental function. Such multi-parameter models are demonstrating superior predictive performance compared to single markers. However, challenges remain in standardization, reproducibility, and cost-effectiveness, which currently prevent large-scale clinical implementation [9,69,72].

### 3.1. Classical Glycemic Markers

FPG: Widely used; FPG in the first trimester shows moderate predictive power with area under the curve (AUC) between ~0.630 and 0.738 (sensitivity 64–79%, specificity 56–59%) when thresholds range from 81 to 88.5 mg/dL [73,74,75,76];HbA_1_c: First-trimester cut-offs (5.33–5.45%) deliver an AUC of 0.809–0.84; in second-trimester, levels (5.45–5.7%) maintain good predictive performance, with an AUC of 0.826–0.848 [76];Insulin-derived indices: Calculated biomarkers such as secretory capacity of pancreatic β-cells (SPINA-GBeta) (a computational estimate of maximal β-cell insulin output) outperform homeostatic model assessment of β-cells function (HOMA-Beta) in estimating beta-cell function and may provide additional insight into early GDM pathophysiology [77].

### 3.2. Emerging Molecular Biomarkers

#### 3.2.1. Placental-Derived Exosomes and Proteomic and Metabolomic Markers

Exosome proteomics and transcriptomics analyses reveal distinct signatures in pregnant women who develop GDM compared to controls. Proteins isolated from plasma- and placenta-derived exosomes show differential abundance in pathways related to insulin signaling, inflammation, and mitochondrial function [62,63,78];Advances in exosome isolation and platform sensitivity are making these vesicular biomarkers more practical for early detection [79].Metabolomic profiling, particularly using targeted acylcarnitines (e.g., C5 isovalerylcarnitine and C5:1 tiglylcarnitine), enabled a predictive model with an AUC of 0.934 for GDM detection before 18 weeks of gestation [78];Combined proteomic/metabolomic fingerprinting studies identify networks of proteins and metabolites that correlate with GDM pathogenesis and postpartum metabolic risk [70,80].

#### 3.2.2. Immune and Cytokine Biomarkers

A recent first-trimester panel including total immunoglobulin G (IgG), total IgM, IL-7, anti-phosphatidylserine IgG immune complexes, and IL-15 achieved AUC of 0.906 (sensitivity 75%, specificity ~95%) in In Vitro Fertilization (IVF) pregnancies. These immune markers reflect early immunometabolic dysregulation and may add predictive power beyond classical glycemic indices [81].

#### 3.2.3. Potential Risks and Roles of Chemerin in GDM

Chemerin, an adipokine, has a complex and important role in the development of GDM, affecting the mother and baby by impacting metabolic, inflammatory, and genetic processes. Elevated circulating chemerin levels serve as a key marker for GDM in mothers, a finding supported by a meta-analysis that demonstrated significantly higher concentrations in GDM cases compared to normoglycemic controls. This link is especially significant among women under 30 and Asian populations. The pathogenic mechanism involves chemerin acting as a pro-inflammatory cytokine, which promotes insulin resistance, a key feature of GDM, by disrupting glucose uptake, influencing pancreatic beta-cell function, and correlating with other inflammatory markers associated with GDM [82].

Besides its systemic effects, chemerin also directly influences placental health by increasing lipid buildup when levels are excessive, disrupting the differentiation and growth of essential trophoblast cells, and impairing overall placental development [83]. Genetic predisposition also influences risk, with specific single-nucleotide polymorphisms (SNPs) in the chemerin gene (*rs4721*) interacting with other variants potentially heightening GDM risk. This may occur through abnormal chemerin secretion and disrupted glucose-lipid metabolism [82,84]. These maternal dysfunctions directly pose risks to the newborn. Higher maternal chemerin levels are associated with increased fetal growth, greater birth weight, and a higher likelihood of large-for-gestational-age infants, which is a common complication of GDM [82]. This effect occurs via the placenta, where chemerin’s control of fatty acid oxidation is essential for the healthy development of trophoblast cells that support the fetus [83]. While changed fetal growth is the main risk, animal studies indicate other possible effects, such as chemerin accumulation potentially leading to fetal cognitive issues or, alternatively, aiding in neutralizing placental oxidative stress. This suggests a complex role that warrants further research [82,84].

### 3.3. Clinical Applicability and Limitations

Timing is crucial, as many molecular markers can be detected in the first trimester, providing an early opportunity for risk assessment and preventive measures, unlike OGTT, which is performed later [73,85];Multivariable models combining conventional risk factors with panels of cardiometabolic biomarkers show improved discrimination for GDM. For example, full prediction models yielded an AUC of 0.842 and a proportion of cases followed (PCF) of 0.695, in early pregnancy, 10 to 14 gestational weeks, compared to an AUC of 0.720 and PCF of 0.491 for the model using only conventional risk factors [72];Key barriers include: deciding on a selective or universal screening, assay standardization (especially for miRNA and exosome quantification), cost, and the need for prospective validation in diverse populations. Clinical uptake will depend on reproducible thresholds, ease of sampling, and cost-effectiveness compared to current universal OGTT screening [86,87,88].

In Table 3, we have summarized the main implications described previously of practical biomarkers in GDM to provide a holistic overview.

## 4. Clinical Management

Effective clinical management of GDM requires an integration of timely diagnosis, lifestyle modification, pharmacologic therapy when indicated, and innovative molecularly informed strategies. As GDM is both a metabolic and molecularly heterogeneous condition, management strategies increasingly aim to move beyond glycemic control to also address underlying pathways such as inflammation, oxidative stress, and microbiome dysbiosis [10,93,94].

### 4.1. Diagnostic Approaches and Lifestyle Interventions

The diagnosis of GDM remains grounded in OGTT, with the IADPSG criteria widely adopted: fasting plasma glucose ≥ 92 mg/dL, 1-h ≥ 180 mg/dL, or 2-h ≥ 153 mg/dL following a 75 g glucose load. While these thresholds are highly sensitive, their specificity varies across populations, and controversies persist regarding overdiagnosis in certain ethnic groups and the lack of early-pregnancy markers [95]. Furthermore, conventional OGTT does not capture molecular heterogeneity of GDM subtypes, including those dominated by β-cell dysfunction versus those driven primarily by insulin resistance [33,40].

Nutritional therapy is the cornerstone of management. Controlled carbohydrate intake, low glycemic index foods, and adequate protein and fiber reduce postprandial glycemic excursions and modulate insulin secretion. Molecularly, nutrition and dietary supplements might attenuate activation of the nuclear factor kappa B (NF-κB) pathway, lower circulating TNF-α, and improve adiponectin secretion, thereby reducing systemic inflammation [22,94,96].

Exercise improves maternal insulin sensitivity by enhancing GLUT4 translocation in skeletal muscle and activating AMPK, which promotes fatty acid oxidation and decreases hepatic gluconeogenesis. Regular moderate activity also reduces oxidative stress by upregulating superoxide dismutase and catalase in placental tissue. Combined lifestyle interventions have consistently shown a reduction in the need for pharmacological therapy, highlighting their molecular and clinical efficacy [22,97,98].

### 4.2. Pharmacological Therapies

Insulin remains the gold standard therapy when lifestyle interventions fail. Due to its large molecular size, it does not cross the placenta, minimizing direct fetal exposure. Insulin therapy primarily corrects maternal hyperglycemia without directly modulating inflammatory or oxidative stress pathways. However, its need for injections, risk of hypoglycemia, and maternal weight gain remain barriers [22,99,100,101].

Metformin is now widely used as a first-line or adjunct therapy in GDM [99]. Mechanistically, metformin activates AMPK, reduces hepatic gluconeogenesis, enhances insulin receptor phosphorylation, and suppresses ROS generation [22,102]. It has also been shown to reduce maternal weight gain and the incidence of large-for-gestational-age neonates [103]. Despite these benefits, metformin readily crosses the placenta, and emerging animal data suggest potential long-term effects on offspring brain development, and it may cause fetal programming, harming metabolic health in later life [104].

Glyburide (Glibenclamide) stimulates maternal pancreatic β-cell insulin secretion via ATP-sensitive potassium channel inhibition. However, it crosses the placenta more readily than metformin, exposing the fetus to pharmacologic activity. Clinical studies have reported higher rates of neonatal hypoglycemia and macrosomia compared to insulin or metformin, limiting its widespread adoption [22,103,105].

Consensus favors insulin as the safest pharmacologic choice, with metformin an effective alternative when patient preference or cost are considered. Glyburide remains a less favored option due to fetal safety concerns [106].

### 4.3. Emerging and Molecularly Targeted Therapies

Given the central role of oxidative stress and systemic inflammation in GDM pathophysiology, several nutraceuticals and supplements have been tested. Magnesium–zinc–calcium–vitamin D co-supplementation reduced circulating high-sensitivity C reactive protein (hsCRP), TNF-α, and malondialdehyde, while improving antioxidant enzyme activity [22,107,108]. Polyphenolic compounds such as curcumin, punicalagin, and naringenin inhibit NF-κB signaling and enhance mitochondrial biogenesis in placental and adipose tissues, offering mechanistic promise [22,109,110,111].

Gut dysbiosis in GDM is characterized by depletion of short chain fatty acid (SCFA)–producing bacteria and expansion of pro-inflammatory Gram-negative species. This promotes insulin resistance via lipopolysaccharide–Toll-like receptor 4 (TLR4) signaling pathways [112]. Probiotic and synbiotic interventions have demonstrated improvements in fasting glucose, HOMA-IR, and inflammatory cytokines, though results vary depending on the bacterial strains used and the dosage administered [113,114,115,116].

New anti-inflammatory compounds, such as NLR family pyrin domain containing 3 (NLRP3) inflammasome inhibitors–1,3,4-oxadiazol-2-one derivative 5 (INF200), have shown notable improvements in glucose tolerance and systemic inflammation in preclinical models of diet-induced metabolic syndrome. Although not yet examined in GDM, these agents underscore the potential of targeted molecular therapies [117,118,119].

Artificial intelligence (AI)–driven closed-loop insulin delivery systems are being explored in type 1 and type 2 diabetes, and their adaptation to pregnancy could provide more precise glycemic control. Reinforcement learning–based insulin titration has shown promising results in simulation models. Translating such systems to GDM may improve maternal and fetal outcomes while reducing the burden of frequent monitoring [120,121,122,123].

## 5. Maternal and Fetal Outcomes

GDM conveys substantial short- and long-term risks for mothers and offspring. Understanding the molecular underpinnings helps inform how these complications emerge and persist [8,124,125].

### 5.1. Short-Term Outcomes

Hyperglycemia in pregnancy leads to fetal hyperinsulinemia and excessive growth, resulting in macrosomia (birth weight ≥ 4000 g), which occurs 2–3 times more often in GDM compared to normoglycemic pregnancies [8,10]. Macrosomia increases risks for shoulder dystocia, Erb’s palsy, neonatal hypoglycemia, hyperbilirubinemia, and admission to neonatal intensive care [8,126,127].

After delivery, infants of diabetic mothers may develop neonatal hypoglycemia due to lingering hyperinsulinemia, along with polycythemia and electrolyte disturbances (hypocalcemia, hypomagnesemia) and increased risk of respiratory distress from delayed lung maturation [128,129].

Women with GDM face a heightened risk of preeclampsia and gestational hypertension. Placental dysfunction driven by oxidative stress, inflammation (elevated TNF-α, leptin, resistin), and endothelial disturbances contribute to increased vascular resistance and hypertensive complications [130,131,132].

### 5.2. Long-Term Outcomes

A history of GDM is one of the strongest predictors of later type 2 diabetes. A meta-analysis of over 675,000 women showed more than a seven-fold increased risk of developing type 2 diabetes in time for the women who developed GDM, compared to women with a normoglycemic status in pregnancy [133,134]. Beyond diabetes, women with prior GDM exhibit increased rates of metabolic syndrome, hypertension, dyslipidemia, and cardiovascular events at younger ages. Prospective cohorts and pooled analyses report up to two-fold or greater elevated risk of cardiovascular disease in this population [27,134,135]. The American Heart Association recognizes prior GDM as a female-specific risk factor for future cardiovascular disease [136,137].

Emerging epidemiological studies link a history of GDM to increased risk of microalbuminuria and chronic kidney disease later in life, even controlling for progression to diabetes [138,139,140].

Intrauterine exposure to hyperglycemia and altered nutrient milieu programs the fetus toward enhanced adiposity, insulin resistance, and risk for childhood and adult obesity and type 2 diabetes, a phenomenon coined metabolic imprinting and consistent with the developmental origins of health and disease hypothesis (DOHaD) [141,142,143]. Longitudinal and mechanistic studies show that maternal GDM and overnutrition increase the risk of metabolic disorders in offspring later in life [125,141].

A recent extensive pooled analysis, around 56 million pregnancies, found that maternal diabetes, including GDM, was associated with a higher risk of neurodevelopmental disorders (1:28 risk ratio), including autism (1:25 risk ratio), attention deficit hyperactivity disorder (ADHD) (1:30 risk ratio), and intellectual disabilities (1:32 risk ratio). While causality is still under investigation, experts emphasize the need for glycemic control to mitigate these risks [144].

### 5.3. Epigenetic and Transgenerational Effects: Integration of Molecular Insights into Outcomes

Molecular evidence suggests that prenatal hyperglycemia triggers epigenetic modifications in key metabolic genes, such as DNA methylation changes and altered microRNA expression. These changes may persist after birth, influencing long-term gene expression and metabolic phenotype in offspring [68,69,145]. Genetic factors potentially influencing GDM risk include SNPs in genes such as *TCF7L2* and succinate receptor 1 (*SUCNR1*), which are linked to changes in insulin secretion and placental endothelial function. These inherited variants operate independently of epigenetic modifications, another regulatory layer important in disease development susceptibility [146,147,148]. Recent studies have begun to link specific molecular biomarkers with clinical endpoints. Elevated placental succinate and *SUCNR1* expression are associated with endothelial proliferation and may underlie placental dysfunction and fetal overgrowth in GDM [147]. Circulating microRNAs and exosomal markers measured during pregnancy are being evaluated for predictive power for GDM onset and the likelihood of macrosomia or future metabolic disease [57,149].

Figure 2 illustrates the clinical consequences and long-term outcomes of GDM.

## 6. Future Perspective

Research into GDM is rapidly expanding, driven by advances in omics technologies, computational modeling, and precision medicine initiatives. Bridging molecular insights with clinical care represents the next frontier in reducing both immediate and long-term complications of GDM [150,151].

### 6.1. Advances in Omics Technologies

High-throughput omics platforms have transformed the ability to characterize GDM at the genomic, transcriptomic, proteomic, metabolomic, and epigenomic levels. Genome-wide association studies (GWAS) have identified SNPs in genes such as *TCF7L2*, *MTNR1B*, and *KCNQ1*, which are linked to impaired insulin secretion and elevated GDM risk [152,153].

Metabolomics has provided new candidate biomarkers—such as altered amino acid, lipid, and bile acid profiles—that show predictive potential for GDM before clinical diagnosis. Integrating multi-omics datasets may allow for precise molecular subtyping of GDM, distinguishing women with predominant insulin resistance versus β-cell dysfunction [9,154].

### 6.2. Precision Medicine, Risk Stratification, and Novel Therapeutic Targets

One of the most promising directions is the translation of molecular findings into precision medicine approaches. Predictive models that integrate clinical risk factors with genetic and epigenetic markers have shown improved accuracy in forecasting GDM development and adverse outcomes. Personalized interventions such as tailoring dietary, lifestyle, and pharmacologic therapies according to molecular subtypes could improve glycemic control and treatment strategies [125,155].

MicroRNA and exosomal profiling are particularly attractive for early screening, as they can be measured in maternal plasma in the first trimester and may predict not only GDM onset but also downstream outcomes such as macrosomia or preeclampsia [9,61,149].

Beyond conventional therapies (insulin, metformin, glyburide), molecular research highlights new avenues for intervention [22,111,156]:Anti-inflammatory and antioxidant therapies: Given the role of cytokine-driven inflammation and oxidative stress in GDM pathophysiology, agents targeting NF-κB signaling, ROS scavenging, or mitochondrial protection are under investigation;Gut microbiome modulation: Dysbiosis in GDM pregnancies suggests potential benefit of probiotics, prebiotics, and dietary interventions to restore microbial diversity and improve metabolic outcomes;Epigenetic therapeutics: While still experimental, strategies that target DNA methylation or histone modification pathways could theoretically modulate fetal programming effects of maternal hyperglycemia.

### 6.3. Digital Health, Predictive Analytics, and Ongoing Challenges

Digital health technologies are increasingly integrated into GDM management. Continuous glucose monitoring (CGM) and wearable sensors provide real-time metabolic profiling, which can be combined with machine-learning models for individualized prediction of glycemic excursions. Mobile health platforms are also enhancing adherence to diet and exercise regimens, with early trials suggesting improved outcomes in glucose control and maternal satisfaction [8,157].

Despite progress, several challenges limit the clinical translation of molecular insights. GDM arises from diverse mechanisms (insulin resistance, β-cell dysfunction, inflammation), complicating biomarker validation. Cost, scalability, and ethical considerations (e.g., genetic testing during pregnancy) remain major obstacles to adopting omics-based diagnostics in clinical practice [9,27,69,158,159].

### 6.4. Toward Integrated Care

Future research must bridge molecular and clinical domains through large, prospective, multi-ethnic cohorts that integrate omics profiling, digital health data, and clinical endpoints. Such efforts could enable the development of clinically actionable molecular signatures and lay the foundation for integrated care models, where early prediction, personalized intervention, and long-term follow-up are standard practice [160].

The future of GDM research lies in leveraging omics technologies, precision medicine, and digital health tools to predict, prevent, and personalize care. Advances in biomarkers, therapeutic targeting of inflammation, oxidative stress, and microbiome modulation, along with machine-learning powered predictive models, hold promise. Yet, successful translation will require rigorous validation, equitable research across populations, and integration of molecular data into accessible clinical workflows [151].

## 7. Conclusions

GDM represents a unique metabolic disorder that emerges in pregnancy but carries lasting implications for maternal and offspring health. Over the past decade, significant progress has been made in unraveling the molecular mechanisms underlying GDM, including the interplay of insulin resistance, β-cell dysfunction, inflammation, oxidative stress, and epigenetic programming. These molecular insights have enriched our understanding of why some women develop GDM and why its effects extend far beyond pregnancy. Figure 3 illustrates a comprehensive overview of GDM.

GDM remains a significant health challenge. While oral glucose tests are standard, they have limitations, emphasizing the need for earlier, less invasive biomarkers. Advances in genomics and metabolomics suggest circulating microRNAs, placental exosomes, and metabolites as promising options. Incorporating these could enable early detection, risk assessment, and personalized treatments. GDM management mainly involves lifestyle, insulin, and oral drugs such as metformin and glyburide. Recognition of inflammation, oxidative stress, and gut microbiota’s roles introduces new therapies beyond glucose control, potentially improving outcomes and reducing long-term risks for mothers and offspring. GDM has significant consequences. Mothers face risks such as preeclampsia and other maternal complications; newborns are vulnerable to macrosomia, hypoglycemia, and respiratory distress. Long-term, mothers face higher risks of type 2 diabetes, cardiovascular disease, and kidney issues, while children are prone to obesity, diabetes, and neurodevelopmental disorders. Epigenetic mechanisms play a key role in transmitting these risks across generations, emphasizing the importance of early intervention.

The future of GDM research and care hinges on integrating molecular discoveries with clinical management. Precision medicine, driven by omics and digital health, can shift from a generic approach to personalized strategies based on each woman’s unique molecular and clinical profile. Challenges include disease heterogeneity, the need for diverse cohorts, and barriers to cost-effective real-world implementation. GDM should be seen not just as a transient pregnancy complication but as a window into future health for mother and child. Combining molecular insights with innovative clinical strategies can reduce the burden of GDM. Progress requires collaboration among scientists, clinicians, public health experts, and policymakers. Integrating science with translational and clinical research is vital for prevention, better maternal–fetal outcomes, and breaking the cycle of intergenerational disease risk.

## Figures and Tables

**Figure 1 ijms-26-08719-f001:**
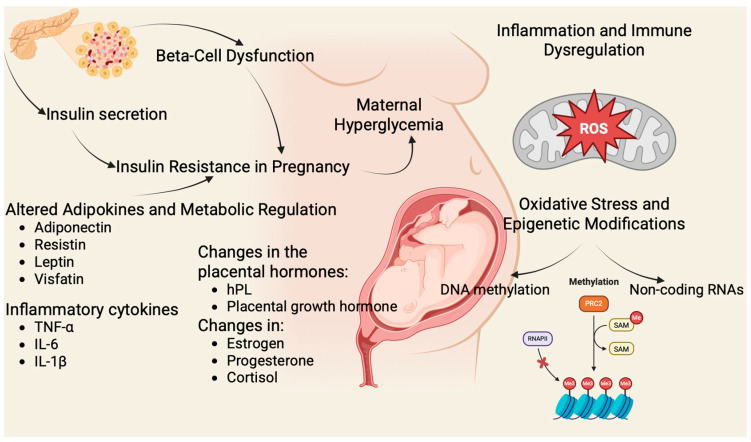
Molecular Mechanisms in GDM. (Figure created in BioRender. Assani, M. (2025) https://BioRender.com/0xnbu0m). Pregnancy-induced insulin resistance is driven by placental hormones (human placental lactogen (hPL), placental growth hormone, estrogen, progesterone, cortisol) and inflammatory cytokines (tumor necrosis factor α (TNF α), interleukin 6 (IL-6), interleukin 1β (IL-1β)) [10,20]. Inadequate β-cell adaptation due to mitochondrial dysfunction, and impaired incretin signaling leads to hyperglycemia. Dysregulated adipokines (↓ adiponectin, ↑ leptin, resistin, visfatin), oxidative stress, and epigenetic modifications (deoxyribonucleic acid (DNA) methylation, non-coding ribonucleic acids (RNAs)) further exacerbate disease and contribute to adverse maternal and fetal outcomes [10,21].

**Figure 2 ijms-26-08719-f002:**
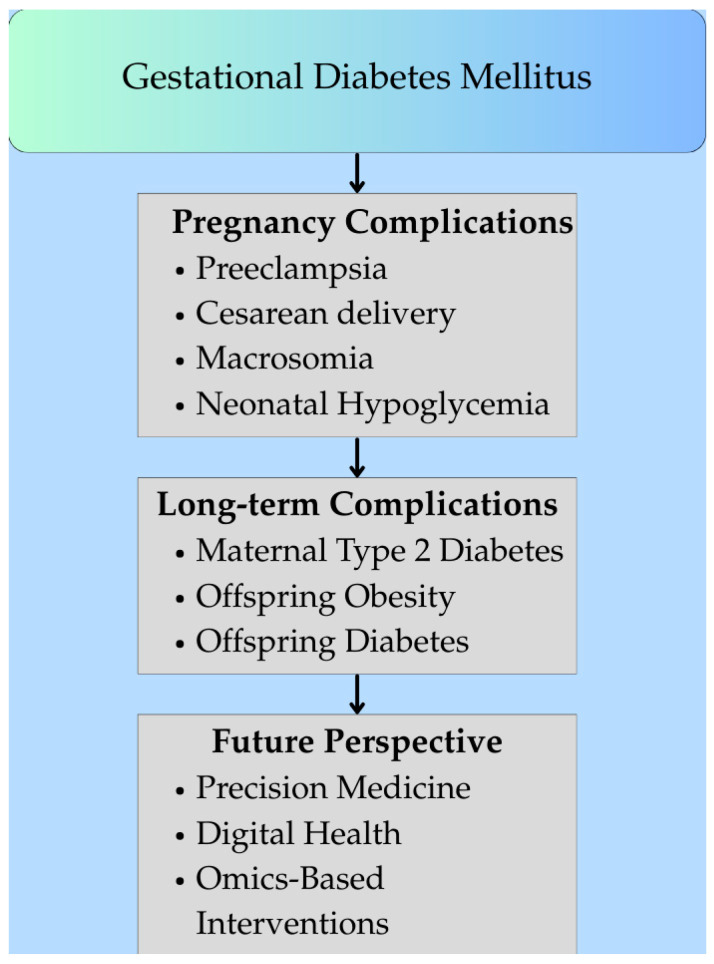
Clinical consequences and long-term outcomes of gestational diabetes mellitus (GDM). (Figure created in Canva, https://www.canva.com). The figure outlines GDM complications. During a pregnancy with GDM, complications such as preeclampsia, cesarean delivery, and neonatal issues (e.g., hypoglycemia) may occur. Long-term, women are more prone to type 2 diabetes, while offspring face obesity and diabetes. Future approaches focus on using precision medicine, omics, and digital health for early detection, risk assessment, and prevention.

**Figure 3 ijms-26-08719-f003:**
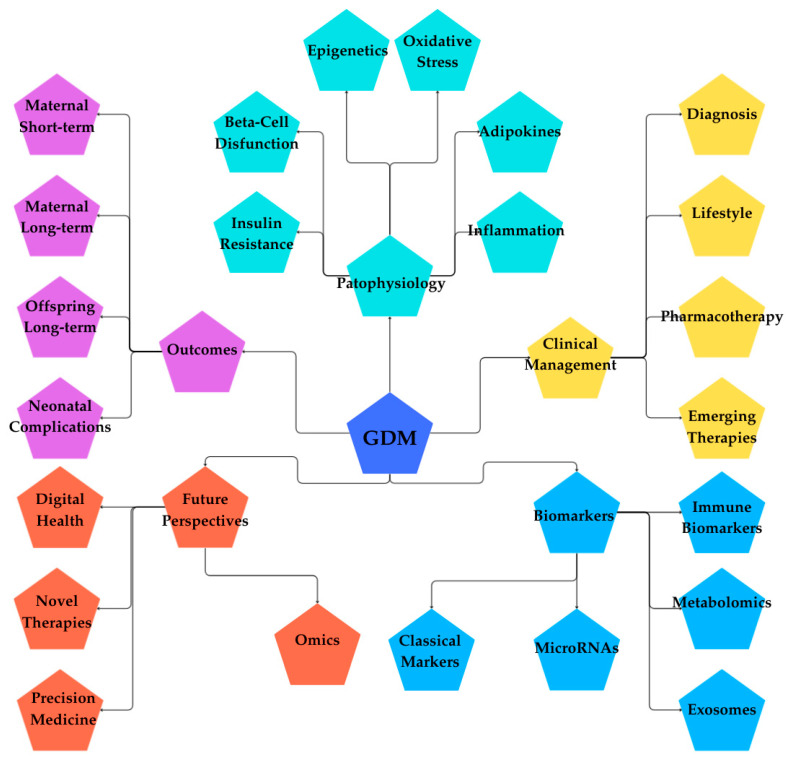
Mind map of GDM. (Figure created in Canva, https://www.canva.com). The mind map provides a holistic overview of GDM, centered on five major domains: Clinical Management, Biomarkers, Pathophysiology, Outcomes, and Future Perspectives. Clinical Management covers diagnosis, lifestyle interventions, pharmacotherapy, and emerging therapies. Biomarkers include metabolomics, exosomes, microRNAs, classical markers, and immune biomarkers. Pathophysiology explores the mechanisms behind GDM, such as epigenetics, oxidative stress, adipokines, inflammation, beta-cell dysfunction, and insulin resistance. Outcomes highlight maternal short- and long-term risks, neonatal complications, and offspring’s long-term health impacts. Finally, Future Perspectives point toward omics, precision medicine, novel therapies, and digital health innovations. This overview demonstrates the complexity of GDM by combining clinical practice, biological mechanisms, patient outcomes, and future research directions into a unified, comprehensive model.

**Table 1 ijms-26-08719-t001:** Brief description of pathophysiology and molecular mechanisms underlying GDM [2,10,12,18,19,20,21,67,68].

Mechanism	Key Drivers	Consequences
Insulin Resistance	Placental hormones (e.g., hPL); cytokines (e.g., TNF-α, IL-6)	Reduced PI3K/Akt signaling; impaired GLUT4-mediated glucose uptake; systemic insulin resistance
β-cell Dysfunction	Oxidative stress, glucolipotoxicity, ER stress, impaired autophagy, ROS accumulation	Loss of insulin secretion; β-cell apoptosis; inadequate compensation for pregnancy-induced resistance
Inflammation	Adipose tissue macrophages; TNF-α; IL-6; JNK/IKK signaling	Amplified insulin resistance; placental inflammation; metabolic dysregulation
Adipokines	↓ Adiponectin; ↑ Leptin; ↑ Resistin; impaired AMPK/GLUT4 signaling	Worsened insulin resistance; chronic inflammation; reduced glucose uptake
Oxidative Stress	Mitochondrial dysfunction; excessive ROS; impaired ATP production	Vicious cycle of metabolic impairment, inflammation, and persistent insulin resistance
Epigenetic/ Genetic	*TCF7L2*, *IRS1*, *MTNR1B* polymorphisms; DNA methylation; histone modifications; microRNAs	Altered gene expression; impaired insulin signaling

hPL: human placental lactogen; TNF-α: tumor necrosis factor α; IL-6: interleukin 6; PI3K/Akt: phosphatidylinositol 3-kinase (PI3K)/protein kinase B; GLUT4: glucose transporter type 4; ER: endoplasmic reticulum; ROS: reactive oxygen species; JNK/IKK: Jun N-terminal Kinase/I-kappa-B Kinase; AMPK: adenosine monophosphate (AMP)-activated protein kinase; DNA: deoxyribonucleic acid; ATP: adenosine triphosphate; microRNAs: micro ribonucleic acids; *TCF7L2*: transcription factor 7-like 2; *IRS1*: insulin receptor substrate 1; *MTNR1B*: melatonin receptor 1B protein; ↓: decreases; ↑: increases.

**Table 2 ijms-26-08719-t002:** Biomarkers in GDM [69,70,71].

Molecular Biology	Biomarkers	Associated Conditions
Genomics, Metabolomics, Proteomics	Circulating microRNAs	Impaired insulin action, beta-cell stress, Inflammation, Oxidative imbalance
	Placental-derived exosomes	
	Metabolic signatures	

**Table 3 ijms-26-08719-t003:** Summarization of practical biomarkers in GDM [2,10,21,62,81,89,90,91,92].

Biomarker Type	Examples	Advantages	Limitations
Classical glycemic markers	HbA1c, FPG	Widely available, standardized	Modest sensitivity/specificity
MicroRNAs	Exosome miRNA panels	Mechanistic insight, early marker	Laboratory complexity, standardization
Proteomics/Metabolomics	Acylcarnitines, exosomal proteins	Predictive power	Costly, specialized platforms
Immune biomarkers	Immunoglobulins (IgG, IgM), Interleukins (IL-7, IL-15)	Reflect immunometabolism	Validation needed, narrow cohorts
Adipokines	Leptin, Adiponectin, Visfatin, Resistin	Mechanistic relevance	Predictive ability

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
