# Peer review of "From Molecular Insights to Clinical Management of Gestational Diabetes Mellitus—A Narrative Review"

_ijms, 2025, doi:10.3390/ijms26178719_

Round 1

Reviewer 1 Report

Comments and Suggestions for Authors

The authors aim to summarize the current literature on the pathogenicity, molecular diagnosis (including molecular biomarkers), and current management approaches for gestational diabetes mellitus (GDM). They also discuss the future potential of molecular biomarkers and novel therapeutic targets related to GDM. Additionally, the authors address the limitations of current clinical practices in utilizing these biomarkers and novel drugs.

Overall, the review is well-written, comprehensive, and provides a thorough summary of recent articles in the field. However, I have a few comments that should be addressed before proceeding further:

  1. The reported GDM prevalence is outdated; please update this information in both the main text and the introduction.
  2. Focusing on Romania seems unnecessary since the review targets an international audience. I recommend deleting lines 51-52.
  3. In Figure 1, what does "hPL" refer to? Please clarify in both the main text and the figure caption.
  4. In Figure 1, should the first organ be the pancreas? If so, please make this clearer.
  5. Line 97: Please rephrase “the beta cells compensate for what?” for better clarity.
  6. Kindly review the spelling in all figures and correct any errors.
  7. The authors might consider discussing the potential risks and roles of chemerin in GDM pathogenicity, particularly with respect to the mother and the newborn.
  8. Illustrations could be helpful in explaining the pathogenicity process, especially insulin resistance.
  9. Please provide more detailed molecular mechanistic explanations for the genetic risk factors, particularly in Section 2.4 (Epigenetic and Genetic Factors, MicroRNAs).
  10. It would be beneficial to cite recent references wherever possible. Please update references 20, 21, 25, 27, 32, 36, 43, 49, 51, 60, and 137 as appropriate.

Author Response

Thank you for your thoughtful and detailed review. Your comments were clear, constructive, and very helpful in identifying areas for improvement. We appreciate the time and effort you invested in reading closely, pointing out strengths and weaknesses, and raising points we had not considered. Your feedback will help strengthen the work and ensure it reaches a higher standard. Kindly point out any additional details that might have been overlooked, or if there are still any of the suggestions that need to be developed further.

Comment 1:

The reported GDM prevalence is outdated; please update this information in both the main text and the introduction.

Response 1: We revised to the prevalence of IDF Diabetes Atlas: Estimation of Global and Regional Gestational Diabetes Mellitus Prevalence for 2021 by International Association of Diabetes in Pregnancy Study Group's Criteria, January 2022, 10.1016/j.diabres.2021.109050. We kindly ask for better support, if it is still not ok.

Comment 2:

Focusing on Romania seems unnecessary since the review targets an international audience. I recommend deleting lines 51-52.

Response 2: We removed the lines.

Comment 3:

In Figure 1, what does "hPL" refer to? Please clarify in both the main text and the figure caption.

Response 3: It is now clarified. (human placental lactogen)

Comment 4:

In Figure 1, should the first organ be the pancreas? If so, please make this clearer.

Response 4: Figure is now revised.

Comment 5:

Line 97: Please rephrase “the beta cells compensate for what?” for better clarity.

Response 5: The sentence was now rephrased, please check.

Comment 6:

Kindly review the spelling in all figures and correct any errors.

Response 6: Updated and revised. Should you find other typos, please mention.

Comment 7:

The authors might consider discussing the potential risks and roles of chemerin in GDM pathogenicity, particularly with respect to the mother and the newborn.

Response 7: Please check Section 3: Potential risks and roles of chemerin in GDM.

Comment 8:

Illustrations could be helpful in explaining the pathogenicity process, especially insulin resistance.

Response 8: Updated.

Comment 9:

Please provide more detailed molecular mechanistic explanations for the genetic risk factors, particularly in Section 2.4 (Epigenetic and Genetic Factors, MicroRNAs).

Response 9: We provided more details. If more are still needed, please mention.

Comment 10:

It would be beneficial to cite recent references wherever possible. Please update references 20, 21, 25, 27, 32, 36, 43, 49, 51, 60, and 137 as appropriate.

Response 10: Revised. We tried to replace some of them, and we also provided more recent references in some places, that might validate the less recent references.

Please kindly review the changes made with tracking.

Reviewer 2 Report

Comments and Suggestions for Authors

I have a very good impression of the manuscript I have read The text is well structured, and the processes are described correctly. As a reader, I didn't have enough tables that would reveal a number of statements to remove the need to search for information in primary sources. In general, all the statements are true, but without supporting information, they look like something general applicable to any multifactorial disease.

  1. I believe that adding tables to the statements below would strengthen the article.

Lines 200-203. Advances in molecular biology and “omics” technologies—such as transcriptomics, metabolomics, and proteomics—have revealed novel classes of biomarkers including circulating microRNAs, placental-derived exosomes, and metabolic signatures that reflect impaired insulin action, β-cell stress, inflammation, and oxidative imbalance.

Lines 233-240. 2.2. Placental-Derived Exosomes

Lines 264-268. Model performance: Multivariable models combining conventional risk factors with panels of microbiological, immunological, and metabolic biomarkers show improved discrimination. For example, full prediction models yielded AUC ≈ 0.84 in early pregnancy versus ~0.49 for risk-only models;

  1. This formulation is not very clear within the framework of the chapter epigenetics, since the loci predisposing to the disease are inherited regardless of whether a case of GDM has occurred or not. Perhaps it meant something else, but then it's better to reformulate this phrase. Lines 408-409

Candidate molecular mechanisms include SNPs in genes like TCF7L2 and SUCNR1 associated with insulin secretion or placental endothelial function in GDM

Minor revision

  1. The names of human genes should be written in italics, for example in lines 58 and so on.
  2. Mechanismsunderlying A typo in the line 200

Author Response

We sincerely thank you for your thorough and thoughtful review. Your constructive observations have been invaluable in identifying areas that need improvement. We greatly appreciate the time and care you devoted to assessing our work, and bringing forward perspectives we had not considered. Your feedback will play an important role in improving the quality of the work. Please let us know if there are any further details we may have overlooked or still need clarification.

Comment 1:

I believe that adding tables to the statements below would strengthen the article.

a)Lines 200-203. Advances in molecular biology and “omics” technologies—such as transcriptomics, metabolomics, and proteomics—have revealed novel classes of biomarkers including circulating microRNAs, placental-derived exosomes, and metabolic signatures that reflect impaired insulin action, β-cell stress, inflammation, and oxidative imbalance.

b)Lines 233-240. 2.2. Placental-Derived Exosomes

c)Lines 264-268. Model performance: Multivariable models combining conventional risk factors with panels of microbiological, immunological, and metabolic biomarkers show improved discrimination. For example, full prediction models yielded AUC ≈ 0.84 in early pregnancy versus ~0.49 for risk-only models;

Response 1: Where the information was suitable for transformation into a table, we did it. If more is still needed, we might try adding more.

a) We added a new table.

b) We relocated this part of information.

c) Rephrased for a better understanding.

Comment 2:

This formulation is not very clear within the framework of the chapter epigenetics, since the loci predisposing to the disease are inherited regardless of whether a case of GDM has occurred or not. Perhaps it meant something else, but then it's better to reformulate this phrase. Lines 408-409

Candidate molecular mechanisms include SNPs in genes like TCF7L2 and SUCNR1 associated with insulin secretion or placental endothelial function in GDM

Response 2: Rephrased for clearer understanding.

Comment 3:

The names of human genes should be written in italics, for example in lines 58 and so on.

Response 3: Revised.

Comment 4:

Mechanismsunderlying A typo in the line 200

Response 4: Corrected.

Please kindly review the changes made with tracking.